# Cell Wall–Based Machine Learning Models to Predict Plant Growth Using Onion Epidermis

**DOI:** 10.3390/ijms26072946

**Published:** 2025-03-24

**Authors:** Celia Khoulali, Juan Manuel Pastor, Javier Galeano, Kris Vissenberg, Eva Miedes

**Affiliations:** 1Department of Biotechnology—Plant Biology, Escuela Técnica Superior de Ingeniería Agronómica, Alimentaria y Biosistemas, Universidad Politécnica de Madrid, 28040 Madrid, Spain; khoulali.celia@alumnos.upm.es; 2Biodiversity and Conservation of Plant Genetic Resources—UPM Research Group, Universidad Politécnica de Madrid, 28040 Madrid, Spain; 3Complex System Research Group—UPM, Escuela Técnica Superior de Ingeniería Agronómica, Alimentaria y Biosistemas, Universidad Politécnica de Madrid, 28040 Madrid, Spain; juanmanuel.pastor@upm.es (J.M.P.); javier.galeano@upm.es (J.G.); 4Grupo Interdisciplinar de Sistemas Complejos (GISC), Madrid, Spain; 5Department of Biology, Faculty of Science, University of Antwerp, 2020 Antwerpen, Belgium; kris.vissenberg@uantwerpen.be; 6Department of Agriculture, Hellenic Mediterranean University, 71410 Heraklion, Crete, Greece

**Keywords:** *Allium cepa* L., machine learning, plant growth, cell wall composition, cell wall enzymes, onion epidermis, modeling

## Abstract

The plant cell wall (CW) is a physical barrier that plays a dual role in plant physiology, providing structural support for growth and development. Understanding the dynamics of CW growth is crucial for optimizing crop yields. In this study, we employed onion (*Allium cepa* L.) epidermis as a model system, leveraging its layered organization to investigate growth stages. Microscopic analysis revealed proportional variations in cell size in different epidermal layers, offering insights into growth dynamics and CW structural adaptations. Fourier transform infrared spectroscopy (FTIR) identified 11 distinct spectral intervals associated with CW components, highlighting structural modifications that influence wall elasticity and rigidity. Biochemical assays across developmental layers demonstrated variations in cellulose, soluble sugars, and antioxidant content, reflecting biochemical shifts during growth. The differential expression of ten cell wall enzyme (CWE) genes, analyzed via RT-qPCR, revealed significant correlations between gene expression patterns and CW composition changes across developmental layers. Notably, the gene expression levels of the pectin methylesterase and fucosidase enzymes were associated with the contents in cellulose, soluble sugar, and antioxidants. To complement these findings, machine learning models, including Support Vector Machines (SVM), k-Nearest Neighbors (kNN), and Neural Networks, were employed to integrate FTIR data, biochemical parameters, and CWE gene expression profiles. Our models achieved high accuracy in predicting growth stages. This underscores the intricate interplay among CW composition, CW enzymatic activity, and growth dynamics, providing a predictive framework with applications in enhancing crop productivity and sustainability.

## 1. Introduction

Plant growth is a fundamental biological process that sustains the biosphere and underpins life on Earth, including human survival. The multifaceted growth phenomenon spans multiple spatial and temporal scales, encompassing cellular development, organ morphogenesis, and ecosystem dynamics, with timeframes ranging from seconds to centuries. At the cellular level, growth originates in meristems, where active cell division, followed by cell expansion and differentiation, drives the formation of new cells and organs, facilitating both structural and functional development. Within plant organs, growth exhibits distinct patterns: unidimensional elongation in roots and stems and bidimensional expansion in several leaf cell types, reflecting the diverse strategies plants employ to adapt to their environments. On a macroscopic scale, net biomass accumulation per unit area and time emerges as a critical indicator of plant productivity, directly influencing global agricultural output and food security [1]. The multiscale integration of these processes underscores the biological, ecological, and economic significance of plant growth, highlighting the need for further investigation to develop predictive models of growth processes that can be utilized to optimize agricultural practices.

The plant cell wall (CW) is a dynamic and resilient biological structure that plays a critical role in growth, development, and the response to environmental signals. The structural organization within the CW plays a critical role in dictating plant morphogenesis and directionality of growth [2] and supports the mechanical properties of the CW, ensuring resilience against turgor pressure while allowing for growth [3]. Structurally, the CW surrounding plant cells is composed of the sugar polymers cellulose, hemicelluloses, and pectins, accompanied by proteins and other compounds such as phenolic compounds and minerals [4].

From a nutritional perspective, these CWs primarily provide energy in the form of carbohydrates (4 kcal/g), protein (7 kcal/g), fibers, minerals, and antioxidant compounds [5]. Thus, plant CWs play a crucial role in human diets not only from an energetic (quantitative) perspective, but also from a nutraceutical (qualitative) standpoint. Nutraceutical compounds (NUTCs) are bioactive molecules, such as flavonoids, phenolics, vitamins, and antioxidants, which confer health benefits by reducing the risk of chronic diseases, enhancing cellular defense mechanisms, and supporting metabolic functions [6]. Among the most significant NUTCs are antioxidants, which protect against chronic diseases such as cancer, diabetes, and metabolic syndrome [7]. Dietary antioxidants are primarily classified into four groups: (*i*) vitamins with antioxidant properties, such as ascorbic acid (vitamin C, oranges), alpha-tocopherol (vitamin E, almonds), and beta-carotene (pro-vitamin A in carrots); (*ii*) carotenoids, including lutein (spinach), zeaxanthin (corn), and lycopene (tomatoes); (*iii*) polyphenols, subdivided into flavonoids (berries) and non-flavonoids (green tea); and (*iv*) other bioactive compounds, such as glucosinolates (isothiocyanates, brassicas) and organosulfur molecules (diallyl disulfide, onion) [8]. Fresh fruits and vegetables provide a range of bioactive compounds, including vitamins, minerals, fibers, and bioactive molecules, which are key contributors to the recommended daily intake of NUTC [9].

Metabolically, the CW functions as a dynamic structure that not only regulates growth but also ensures mechanical stability. This functionality is encoded by cell wall enzymes (CWEs), whose spatiotemporal expression patterns during growth orchestrate plants’ response and effective adaptation to both internal and external stimuli [10]. These enzymes modulate both the quantitative and qualitative aspects of the CW, influencing its structural integrity and nutraceutical profile.

CWEs activities regulate the structural integrity of the plant CW and enhance the functional properties of nutraceutical compounds, such as dietary fiber and antioxidants, highlighting their dual significance in plant physiology and human health. Previous studies have established a direct relationship between the sugar content of the tomato CW and the synthesis and accumulation of secondary metabolites, including phenols and flavonoids [11]. Moreover, alterations in cellulose structure may restrict the ability to synthesize and store carotenoids and limit the mobility of other CW components, thereby influencing the accumulation of flavonoids and phenolic acids [12,13]. Beyond their structural roles, carotenoids are pivotal in abscisic acid (ABA) signaling pathways, which regulate key processes in plant growth and development [14]. During fruit ripening, enzymatic degradation and modification of pectins by polygalacturonase (PG) and pectin methylesterase (PME) play a crucial role in modulating the availability and bioavailability of bioactive compounds, such as carotenoids and polyphenols, as well as the accumulation of phenolic compounds [15]. These findings underscore the intricate interplay between CW dynamics and the metabolic pathways governing the accumulation of nutraceutical compounds, offering insights into their broader implications for both agricultural productivity and human nutrition.

In this study, ten key genes encoding CWEs were selected to assess their roles in regulating and maintaining CW composition and properties. Xyloglucan endotransglucosylase/hydrolase (XTH) restructures xyloglucans, and can thus enhance wall extensibility [16], while expansin disrupts hydrogen bonding between cellulose and hemicelluloses, promoting stress relaxation [17]. Galactosidase and fucosidase cleave terminal sugars from polysaccharides [18], altering wall properties and impacting nutraceutical composition. PG degrades pectin [19], affecting cell adhesion, whereas pectate lyase-like (PLL) cleaves pectate [20], modifying wall porosity. PME demethylates pectin [21], increasing cross-linking potential and rigidity [22]. Xylosidase hydrolyzes xylose residues [23], impacting hemicellulose composition and CW plasticity, and glucanase and xylanase cleave cellulose and xylans [24,25], facilitating loosening and growth [26]. Proteomic analyses reveal how differential accumulation of proteins (e.g., β-1,3-glucanase) in the onion epidermis correlates with responses to external signals and developmental adaptations [27].

The concept of plant growth is inherently broad, encompassing an array of interrelated processes that overlap across spatial and temporal scales [1]. This study narrows its focus to growth defined in *sensu stricto*, referring to irreversible structural and expansive processes. To streamline the experimental framework, temporal variability was minimized by selecting the growth of adaxial epidermal cells in onion (*Allium cepa* L.) bulbs as a model system. Single-point analysis of all onion layers captured sequential growth dynamics within the same individual at one moment in time. This model provides valuable insights into the mechanisms of plant growth; however, its relevance to animal systems is highly limited, with applicability confined solely to processes associated with dental growth [28]. The growth model in onion epidermal cells has been previously studied from a physicochemical perspective [29]. This anisotropic growth model, based on the adaxial epidermis of onion, serves as a framework for understanding how the orientation of cellulose microfibrils dictates the direction, though not the extent, of cell expansion. Microfibril orientation shifts between transverse and longitudinal alignments during development, directing cell expansion. Biomechanical studies have demonstrated that CWs extend more readily in the direction perpendicular to the microfibril alignment, underscoring the regulatory role of additional protein factors, such as CWEs, in anisotropic growth [29]. Moreover, cellulose fibers in onion epidermal cells exhibit a bimodal angular distribution, alternating at ±45° relative to the cellular axis, which forms interwoven layered structures enriched with homogalacturonan networks [30]. The dynamic interplay between CW compounds (CWCs), including the alignment of cellulose microfibrils and the remodeling of the pectic matrix, is essential for maintaining anisotropic growth [31]. This intricate coordination highlights the structural and biochemical complexity underlying cell expansion in plant systems. These findings underscore the intricate interplay of mechanical, biochemical, and genetic factors in the regulation of plant growth. However, gaps remain in our understanding, particularly regarding the expression of CWE genes concerning plant growth, the modification of CWCs, and their integration into predictive modeling frameworks.

Machine learning (ML) has revolutionized numerous scientific disciplines by enabling the analysis of complex and high-dimensional datasets that were previously intractable. In the context of biology and agriculture, ML provides powerful tools to identify patterns, generate predictions, and optimize experimental designs, often revealing insights that would otherwise remain hidden [32]. For instance, ML has been successfully applied to predict rust resistance in wild lentil populations, a key trait for in situ conservation strategies. By using ML models, researchers have estimated severity values under current and future climate scenarios based on IPCC models, identifying candidate populations likely to retain resistance despite environmental changes [33]. Particularly, in plant breeding, ML addresses key challenges, specifically in genomics, crop monitoring, and phenotyping [34]. Across various applications, ML algorithms enable precise genotype–phenotype mapping, trait identification, and classification of plant genotypes under diverse conditions, leveraging temporal and spatial data to enhance agricultural productivity [35]. ML methods enhance analytical accuracy by effectively addressing both regression and classification problems, leveraging their capacity to integrate and analyze multi-scale datasets [36]. These approaches underline the role of computational models in optimizing the study of growth patterns, disease resistance, plant responses to abiotic stress, and breeding programs, improving crop yield [37,38]. Large-scale predictive plant growth models linked to global climate change or net biomass production in agriculture have been developed [39]. However, they cannot predict morphogenetic patterns’ spatiotemporal diversity [1].

## 2. Results

### 2.1. Analysis of Adaxial Epidermal Cell Size Using Cell Wall Staining

A comprehensive visual representation of the structural organization and adaxial cellular morphology across distinct layers (S1–S6) and in the basal (B), medium (M), and upper (U) zones can be seen in the onion (*Allium cepa* L.) bulb (Figure 1A), utilized as a CWE-mediated growth model in this study.

In Figure 1A, the left panel illustrates a schematic cross-section of the onion bulb, showing that it is spatially differentiated into six concentric layers (S1–S6), each of which is subdivided into three zones: the basal (B), medium (M), and upper (U) zones. These layers and zones facilitate the study of region-specific CW growth. The right panel (Figure 1B) comprises microphotographs of adaxial epidermal cells from each layer (S1–S6) and zone (B, M, and U) stained with trypan blue. These images simultaneously reveal cell size, shape, and orientation variations within the epidermal layer.

The average size of onion epidermal cells and the histogram distribution of cell size in the adaxial epidermis of *Allium cepa* L., organized across the six epidermal layers and in the three distinct zones within each layer, were extracted from the images (Figure 2).

The box plots in Figure 2A illustrate the mean cell size distributions, and statistical analysis revealed significant differences in average cell size among these three zones in each of the six layers, except between B and U in the S6 layer. In layer S1 (left top panel), cells in the middle zone exhibited the significantly largest median size (S1_M 1636.9% to S6_M), followed by the upper zone (S1_U 553.6% to S6_U) and basal zone (S1_B 305.6% to S6_B), with the latter showing the smallest cell sizes. This result is also consistent across layers S2 to S5, suggesting marked spatial differentiation between zones in layers and between layers. In the S6 layer (the innermost layer), the cell sizes were more similar, with no significant differences between zones B and U, although the size of cells in the M layer was still significantly larger. The number of cells that could be counted in each zone (n) and in each layer, in the fixed microscopic observation area, supported the results shown. The dispersion of cell sizes, as indicated by the interquartile range (IQR) and the distribution of outliers, was also wider in the middle zone in most layers, and particularly pronounced in layers S4 and S5. In contrast, the basal and upper zones showed narrower IQRs and fewer extreme values, implying a more uniform distribution of cell size in these regions.

On the cell size distribution plots (Figure 2B), it is evident that the oldest layers (S1, S2, and S3; red) showed a predominance of larger cells (around 20.000 μm^2^). A shift was observed in the intermediate layers (S4 and S5; blue), suggesting a decrease in the average cell size (around 10.000 μm^2^). Finally, in the youngest layer (S6; green), the distribution reflects the presence of smaller cells (around 2.000 μm^2^), consistent with mitotic activity and limited cell expansion in this layer close to the meristem.

To elucidate the pattern of size variation in epidermal cells across onion layers (S1–S6) in specific zones (B, M, and U), a proportional constant growth model (PCGM) was proposed. Considering the histograms and the medium size of cells through layers and zones, the positive constant growth factor was applied to all the cell sizes in zone B, and the modified histogram (BX) was obtained, which was then compared (blue arrows) to layer M histograms using the Kolmogorov–Smirnov test. In this analysis, we did not reject the null hypothesis (H0) that the distributions are similar across layers, confirming that the distributions are equal, except for layer S6 (Appendix A), where significant differences were detected. Similarly, a negative constant growth factor was applied to the U histograms (designated UX) and compared (green arrows) to the histograms of the M zone. Likewise, it was unable to reject H0, indicating that growth in these zones aligns with a similarly consistent proportional reduction across all layers.

Once the PCGM was statistically confirmed, it was decided to select the middle zone (M) and layers S1, S2, and S6 to model epidermal growth through the onion layers. Therefore, the results of the analysis of CWCs and NUTCs were related to differential expression of CWEs in each layer, which enabled us to describe the epidermal growth of the entire plant through an individual at a single point in time.

### 2.2. FTIR and Biochemical Spectroscopic Analysis

#### 2.2.1. Cell Wall Compounds: FTIR Analysis

The FTIR spectra recorded for each onion epidermal layer identified eleven distinct spectral intervals with statistically significant structural and compositional differences among layers S1, S2, and S6 (Figure 3A, Appendix A). Based on previous research [40,41], these intervals (Ints) were associated with key cell wall components, including Int1 (pectin ring), Int2 (glucan), Int3–Int5 (cellulose), Int6 (pectin-like structures), Int7 (phenols), Int8 (protein amide II), Int9 (protein β-loops), Int10 (aromatic compounds), and Int11 (esterified carboxylic groups of pectin).

Notably, as depicted in Figure 3A, mature layers (S1) exhibited significantly higher signals for cellulose-associated bonds (Int3–Int5), phenolic compounds (Int7), protein β-loops (Int9), aromatic compounds (Int10), and esterified pectins (Int11). These results highlight the pivotal role of cellulose content, phenolic cross-linking, and pectin esterification in enhancing cell wall rigidity and reducing porosity in mature tissues. In contrast, younger layers displayed higher signals for pectin rings (Int1 and Int6), glucans (Int2), and protein amide II (Int8), which are associated with increased wall elasticity and the incorporation of new material to support cell expansion. This distribution reflects dynamic remodeling of the CW during growth and differentiation. It is crucial to acknowledge that FTIR spectroscopy provides a comparative analysis of the chemical bonds associated with CWC but does not quantify their absolute concentrations. Therefore, these findings require validation through biochemical quantification of the relevant compounds.

To further analyze the observed differences, the integral of the curve of the FTIR signals within each interval was calculated, providing a robust metric for characterizing structural and compositional variations across developmental layers. The FTIR data integrals can be used as features for supervised classifier algorithms. We applied several ML algorithms, including Neural Networks, kNN, SVM, Random Forest, Gradient Boosting, and Logistic Regression. Figure 3B presents the classification accuracy (CA) and precision for each model. Among these, SVM and Gradient Boosting achieved the highest CA and precision. Our goal was to predict the layer (target variable) from the integrals of the FTIR data in different intervals (input variables). The output of the SVM and Gradient Boosting algorithms are models of our data that can be represented as a confusion matrix (Figure 3C), where the dataset shows the percentage of correctly and incorrectly classified instances. Note the high performance of these models generated with nine biological replicates, demonstrating that FTIR data can be used to determine the evolutionary state of an onion. Figure 3D shows the contribution of the ranges to the model. It can be observed that Int1 (840–880), corresponding to the pectin ring, contributes most notably (13%), followed by the range Int11 (1720–1760) corresponding to esterified carboxylic groups of pectin, which contributes 7%. The ranges Int7 and Int6 had a lower contribution of 1.5%.

#### 2.2.2. Biochemical Analysis: Cell Wall and Nutraceutical Compounds

The results of the spectroscopic analysis of the CW compounds of onion epidermal cells are shown in Figure 4A. The total sugar content was significantly higher in layer S6 than in both S2 and S1 and between S2 and S1, suggesting that, sequentially, the innermost scale layer (S6 and then S2) stores more soluble sugars. This potentially reflects its role in osmotic regulation and cellular turgor maintenance. Uronic acid levels, indicative of pectin content, were statistically elevated in S6 relative to S1 layer. The total insoluble sugar content, primarily attributed to α-cellulose, was significantly lower in S6 (349.2 μg/mg dry CW) compared to S2 (431.2 μg/mg dry CW) and S1 (495.9 μg/mg dry CW). Also, α-cellulose content in S2 was significantly lower compared to S1. Reducing sugars were found at significantly higher levels in the S6 layer than in S2 and S1, and also between S2 and S1.

Regarding the nutraceutical analysis of the onion epidermis (Figure 4B), a significantly lower antioxidant capacity was observed in layer S6 compared to both S1 and S2. The S2 layer showed a significantly higher level of antioxidants than S1. Similarly, total phenol content was significantly lower in S6 compared to S2 and S1. Furthermore, flavonoid content was significantly lower in S6 than in both the S1 and S2 layers. The S2 layer showed a flavonoid content that was significantly higher than that seen in S1. The total protein content did not vary significantly among the layers.

#### 2.2.3. Machine Learning Models for Layer Classification Using the CWC and NUTC Data

Supervised analysis was performed to apply classification algorithms and obtain predictive models of CW growth and development (Figure 5). These models enabled classification of onion layers (S1, S2, and S6) based on CW and nutraceutical compounds. Seven different classification methods (Orange3) were applied. The tables (Figure 5A,D) indicate the performance metrics, including classification accuracy and precision, for each algorithm. The SVM model outperformed the others, achieving the highest classification accuracy and precision (95.7% for CWC and 82.7% for NUTC) and precision (96.1% and 82.7% respectively). For the CWC-based predictions, the confusion matrix (Figure 5B) revealed that all three layers were predicted with excellence accuracy: 100% accurate for layers S1 and S6 and 85.7% accurate for layer S2. In Figure 5C, it can be seen that the most important features for classification were α-cellulose and total sugars.

The SVM model also demonstrated strong performance for NUTC data (Figure 5D). The confusion matrix (Figure 5E) indicated correct predictions for layer S6 in 96.3% of cases, while layers S2 and S1 were predicted with 77.8% and 74.1% accuracy, respectively. Among the NUTC parameters, antioxidants (Figure 5F) emerged as the most important feature to build the model. Notably, combining all parameters or limiting the dataset to the most significant variables (e.g., cellulose, total sugars, and antioxidants) did not enhance model performance beyond the results achieved in this study. These findings highlight the robustness of the SVM model in integrating CWC and NUTC datasets for precise growth layer classification.

### 2.3. Gene Expression Analysis of Cell Wall Enzymes by RT-qPCR

A total of ten genes encoding CWEs (PME, xylanase, glucanase, xylosidase, PLL, galactosidase, fucosidase, XTH, expansin and PG) were selected, and their expression was evaluated by RT-qPCR (Figure 6). PME expression was significantly higher in the S1 layer compared to S2 and S6 and also between the S2 and S6 layers. The S1 layer exhibited significantly higher galactosidase expression compared to S2 and S6. Although no significant differences were observed among the S1, S2, and S6 layers in xylanase expression, it was slightly higher in S1, with no statistically relevant changes. Fucosidase expression was significantly higher in S6 compared to S1 and S2. Glucanase expression was significantly higher in S1 compared to S2 and S6, followed by S2, then S6. XTH expression was significantly lower in S2 than in both S1 and S6, although no differences were found between S1 and S6. Similar observations were made regarding xylosidase expression. Likewise, the S2 layer showed significantly lower expression of PLL compared to the S1 and S6 layers. Expansin expression was significantly higher in S2 compared to S1. Finally, the S6 layer exhibited significantly higher expression of PG compared to S2.

Correlations among different CWC and NUTC features and gene expression were assessed (Table 1) using unpaired data, because the samples used in biochemical techniques are different from the samples used in gene expression techniques. To address the imbalance in the dataset comprising CWC/NUTC features and gene expression levels, bootstrap methods were employed for hypothesis testing, standard error estimation, and confidence interval determination. The results of these correlations are illustrated in Appendix A. 

Correlation analysis revealed distinct expression patterns for fucosidase and PME, highlighting their differential roles in regulating cell wall metabolism. Fucosidase expression exhibited a strong positive correlation with CWC features, including total sugars and reducing sugars, with correlation coefficients of 0.96 and 0.97, respectively. Conversely, fucosidase displayed a significant negative correlation with α-cellulose (−0.84) and NUTC features, such as total antioxidants and total phenols (−0.91 and −0.88, respectively). In contrast, PME expression demonstrated an opposite pattern, showing a strong negative correlation with total sugars and reducing sugars (−0.95 for both), while positively correlating with α-cellulose (0.88) and total phenols (0.82). Additionally, glucanase expression showed a notable negative correlation with total antioxidants (−0.87).

## 3. Discussion

Plant growth involves processes occurring across a wide range of spatial and temporal scales, from cellular development to ecosystem-level dynamics [1]. Understanding the mechanisms of growth through the CW is critical for enhancing crop productivity. This study investigated the growth of onion (*Allium cepa* L.) epidermal cells, utilizing ML predictive models to gain insights into the genomic, molecular, structural, and biochemical modifications that govern this process.

The onion epidermis was selected as an ideal model system due to its capacity to provide a simultaneous view of all physiological growth stages. A graphical abstract is provided in Appendix A, illustrating the mathematical analysis performed in this study. Quantification of cell number and size across six layers and three zones revealed a progressive increase in cell size (1637%) from younger to older layers. Notably, cells in the middle zone (M) were significantly larger than those in the basal (B) and upper (U) zones across all layers. These uniform and consistent growth patterns (BX and UX) highlight the precise regulation coordination of growth processes [42,43]. Proteomic studies further confirmed differential protein accumulation across layers, emphasizing the molecular basis for these structural adaptations [27]. For instance, analysis of the lower and upper epidermis of onion scales identified 31 unique proteins. These proteins are mainly involved in pigment synthesis, stress response, and cell division. Notably, chitinase and β-1,3-glucanase were more abundant in the upper epidermis. Moreover, proteins involved in cell division, like β-1,3-glucanase, showed differential accumulation, influencing cell size differences among the layers [27]. These results illustrate the complexity of CW dynamics and their pivotal role in tissue expansion [44,45,46]. Growth regulation is a multifactorial process, integrating mechanical, biochemical, and genetic components, and is further supported by hormonal gradients and developmental patterns within tissues [47,48]. The prevalence of large cells in the middle zones (M) of mature tissues (S1, S2, and S3; 26,750.2 µm^2^, 17,932.3 µm^2^, and 17,547.0 µm^2^, respectively) compared to younger tissues (S4, S5; 8636.9 µm^2^ and 6511.7 µm^2^, respectively), and especially S6 (1540.1 µm^2^), supports anisotropic expansion. This observation underscores structural reorganization of the CW, particularly the alignment of cellulose microfibrils, as a key driver of bulb development [29].

To validate these findings, Fourier transform infrared (FTIR) spectroscopy was employed to analyze the CW. The analysis revealed significant spectral differences across 11 regions corresponding to CWCs, such as cellulose, pectins, and glucans, as well as NUTCs, including phenols, aromatic compounds, and proteins linked to assigned *m/z* values [41]. The FTIR results indicate that layers S1 and S2 contained higher levels of cellulose, esterified pectins, and phenolic compounds compared to S6, which exhibited increased pectins, glucans, and protein amide II groups. These findings align with prior studies demonstrating that younger tissues contribute to CW elasticity and resilience, directly influencing its physical properties [2,10,49]. The integration of CW FTIR spectral data with ML algorithms not only offered precision in layer classification (91.7%), but also revealed the pivotal role of pectin and its esterification in onion development. This integration of FTIR spectral data with structural features effectively highlighted the role of pectin in layer differentiation and growth. This is the first reported application of linear ML models for CW analysis across developmental stages in plant tissues and demonstrates the potential for employing similar methodologies in other plant systems to advance predictive modeling of developmental and physiological processes, ultimately contributing to crop improvement strategies.

To establish a correlation between structural changes and genetic modifications, it is essential to quantify how CWCs and NUTCs evolve during growth, alongside examining components like flavonoids and reducing sugars that were not identified by FTIR. The CWCs, such as total and reducing sugars, along with pectins to a lesser extent, exhibited a significant increase from the outermost, mature layers (S1) to the innermost, younger layers (S6). This trend likely reflects the involvement of soluble sugars in osmotic regulation through turgor pressure, a primary driver of growth in plant tissues [50,51]. Conversely, cellulose content showed a significant, sequential decrease from mature to young tissues (from S1 to S6), consistent with prior research indicating that cellulose deposition reinforces CW structure and provides mechanical protection in mature tissues [52,53,54].

The distribution of NUTCs across onion layers revealed significantly higher levels of antioxidants, phenolics, and flavonoids in mature tissues (S1 and S2) compared to younger tissues (S6), confirming the same behavior as ferulic acid content in onions [55]. This observation underscores the structural role of NUTCs in reinforcing the CW. Compounds like ferulic acid contributes to CW rigidity and resistance to mechanical stress by forming cross-links with pectins and lignins in the outer layers [56,57]. Secondary metabolites such as quercetin mitigates oxidative stress and protect against UV damage, emphasizing their defensive role during growth [58]. In contrast, younger tissues exhibited lower antioxidant activity but higher concentrations of soluble sugars and pectins, suggesting a metabolic focus on biomass accumulation rather than structural reinforcement [46]. These results reflect the adaptive strategies of epidermal layers to meet varying environmental and mechanical demands, aligning with studies linking NUTC gradients to functional tissue differentiation [10,59]. This comprehensive analysis underscores the interplay between CWCs and NUTCs in regulating growth and adaptation, offering new insights into their roles in plant development and stress resilience.

The quantified CWC and NUTC data enabled the application of linear ML models to develop predictive frameworks for determining the growth stage of plant tissues. This approach facilitated the identification of a tissue’s developmental state through the biochemical analysis of only one or two compounds. When integrating all CWC parameters, the SVM model achieved an excellent classification accuracy of 95.7%. By contrast, when NUTCs were analyzed collectively, the best-performing ML model remained SVM, yielding a strong but lower classification accuracy of 82.7%. These findings suggest that CWCs play a more critical role than NUTCs in defining the growth stage of plant tissues. Among the parameters evaluated, cellulose and total sugars emerged as the most influential factors contributing to classification accuracy, followed by antioxidants. This study represents a pioneering effort in quantifying the contributions of CW compounds to tissue growth classification, offering a comparative analysis against NUTCs, including secondary metabolites. The findings highlight the efficacy of ML methodologies, particularly SVM, in synthesizing complex biochemical datasets to accurately classify tissues by growth stages and biomass production. This research provides a robust framework for applying ML in agricultural innovation, paving the way for enhanced crop productivity through precise monitoring and optimization of growth stages [38,39,56,60].

To enable the agronomic applications of this study, it is essential to understand the differential expression of genes responsible for CW metabolism throughout the various growth stages. This research represents the first analysis of ten cell wall enzymes (CWEs) in actively growing tissues. The observed expression patterns revealed distinct responses reflecting structural and functional adaptations during onion bulb development. The first expression pattern (S1 > S2 > S6) demonstrates a significant and sequential increase in PME, glucanase, and galactosidase activity from the youngest tissue (S6) to the most mature layer (S1). PME facilitates the demethylesterification of homogalacturonan (HG) in pectin, enhancing CW rigidity in mature layers [61]. Moreover, the hydrolytic activity of glucanase facilitates wall loosening and expansion by cleaving β-1,4-glucan bonds in cellulose, thereby enhancing wall flexibility [24]. β-Galactosidases hydrolyze galactose residues from complex polysaccharides, such as pectins and hemicelluloses, and are notably expressed in elongation and growth zones [62,63]. These changes support anisotropic expansion and the incorporation of new material via xyloglucan transglycosylation mediated by XTH [64].

The second “V-shaped” pattern (S1 = S6 > S2) was exhibited by the enzymes XTH, xylosidase, and PLL, with a less pronounced effect observed in PG. All of these enzymes showed reduced expression in the intermediate growth layers. The absolute expression of XTH was the highest recorded in this study, likely reflecting the dual enzymatic activities of isoenzymes within the family at different developmental stages [65,66]. In young tissue (S6), hydrolytic isoenzymes appear to increase their expression, cleaving xyloglucan and facilitating wall extension [67,68]. In contrast, mature tissue (S1) exhibits expression of xyloglucan transglycosylase enzymes, which incorporate new oligosaccharides to elongate and strengthen the hemicellulose structural matrix, consequently enhancing cell wall rigidity [41,69]. A similar phenomenon is observed with xylosidase expression, which suggests hydrolytic degradation of hemicellulosic polysaccharides (xylans and xyloglucans), facilitating sugar mobilization during the energy demands of growth and working in conjunction with endoxylanases and galactosidases in mature tissues [70]. Furthermore, layer S2 showed significantly lower PLL expression, suggesting that pectin depolymerization via α-1,4-glycosidic cleavage of galacturonic acid chains from demethylesterified HG is more active in younger tissues (S6) and less active in older tissues (S1). This contributes to a potentially less rigid and more porous structure in these layers [71,72]. PG exhibited low absolute expression, which was significantly higher in younger tissues, potentially indicating galacturonic acid hydrolysis activity in HG [73]. This activity may facilitate CW remodeling in collaboration with pectinases and pectate lyases [74].

The third pattern demonstrated a significant decrease in fucosidase expression as the tissue matured (S6 > S2 = S1), suggesting that, in younger tissues, the release of fucose residues from xyloglucan and glycoproteins may help maintain turgor pressure and mobilize the energy resources necessary for growth [75]. Expansins, with low absolute expression, weaken non-covalent interactions between CWCs, primarily cellulose and xyloglucans. These non-enzymatic proteins displayed a significant reduction in expression from younger to mature tissues (S6 = S2 > S1). This trend indicates that expansin ceases to disrupt non-covalent interactions once cell expansion and growth are completed in the most mature tissue layer (S1) [76]. All these findings align with the critical role of CWEs in maintaining a balance between extensibility and rigidity, which is essential for the sequential growth of plant tissues [46].

Finally, to correlate genetic changes with growth and biomass production, supervised analytical methods, including bootstrap and Pearson correlation, were employed to identify CWC and NUTC parameters associated with CWE gene expression patterns during growth. Pearson correlation coefficients exceeding 0.5 revealed two distinct patterns. PME exhibited robust positive correlations with α-cellulose, antioxidants, and phenolics, but negative correlations with total and reducing sugars. This association underscores the role of PME-mediated HG demethylesterification in mature layers, but not in pectin content, where increased cellulose and antioxidant contents enhance CW rigidity [71,77]. Fucosidase, in contrast, demonstrates a positive correlation with total and reducing sugars while exhibiting a negative correlation with α-cellulose, antioxidants, and phenolics. This enzyme, expressed at approximately 15% of the absolute levels of PME, facilitates the hydrolysis of fucose residues from xyloglucan and glycoproteins in younger layers. In these tissues, total and reducing sugars play a crucial role in maintaining turgor pressure, driving CW extensibility, guiding growth, and providing the energy necessary for these processes [78]. These findings highlight the functional specialization of CWE, where PME enhances rigidity in mature layers, while fucosidase promotes extensibility and energy mobilization in younger layers, highlighting their complementary roles in tissue-specific growth stages and biomass production.

## 4. Materials and Methods

### 4.1. Plant Material

The experimental work was carried out with two similar varieties of onion (*Allium cepa* L. var. Grano and Spring) collected from local markets in Belgium and Spain. The layers (first layer, oldest: S1 to last layer, youngest: S6) of each onion bulb were separated manually and cut into three almost equal portions: the basal zone (B), middle zone (M), and upper zone (U). From each layer, the adaxial epidermis was removed manually and immediately frozen in liquid nitrogen before being stored at −20 °C or −80 °C until further use.

### 4.2. Microscopy Analysis of Cell Morphology

The trypan blue staining method was used to stain the adaxial epidermal cells. A 4–5 mm square piece from each onion bulb layer was incubated for 5 min with trypan blue (Sigma Aldrich, Darmstadt, Germany), washed with water, and observed under the microscope (10×; OPTIKA microscope, Ponteranica, Italy). 

### 4.3. Spectrophotometric and Biochemical Analysis

#### 4.3.1. Cell Wall Compound (CWC) Analysis

Cell wall isolation: One gram of each onion adaxial epidermal layer was collected and placed in glass tubes. Cellulose served as a positive control to verify extraction efficiency. The material was homogenized in ethanol, boiled for 25 min, then filtered and centrifuged at 8500× *g* for 5 min. It was washed twice with ethanol, twice with acetone, and once with methanol:chloroform (1:1, *v:v*). Finally, it was dried at room temperature until a constant dry weight was achieved [79].

AIR (alcohol insoluble residue) cell wall fractionation: Two milligrams of dried purified CW (AIR) preparations was hydrolyzed with 2 M TFA at 121 °C for 1 h and then centrifuged for 5 min. The resulting soluble sugars were stored at −20 °C for quantification. TFA-resistant material was washed twice with water and twice with acetone and dried at room temperature. It was then hydrolyzed in 200 µL of 72% H_2_SO_4_ at room temperature for 1 h, diluted in 0.5 M H_2_SO_4_, hydrolyzed again for 1 h at 121 °C, and stored at −20 °C for quantification of insoluble sugars (cellulose).

FTIR analysis: To study the CW composition of onion epidermis cells, Fourier transform infrared spectroscopy (FTIR) was performed as described in [40] using 5 mm long subapical segments of onion epidermis. A total of 128 interferograms was collected for each sample in transmission mode at 8 cm^−1^ resolution using a Thermo-Electron instrument. All spectra were baseline-corrected and area-normalized (800–1800 cm^−1^) using Thermo Scientific™ OMNIC™ Series Software (https://www.thermofisher.com/order/catalog/product/INQSOF018 (accessed on 18 March 2025)). Eleven intervals where the absorbance of the different layers was significative shifted (*t*-test, *p* ≤ 0.05) were identified: Int1 (pectin ring) = [840–880], Int2 (glucan) = [900–940], Int3 (cellulose) = [1090–1120], Int4 (cellulose) = [1140–1170], Int5 (cellulose) = [1300–1360], Int6 (pectin-like) = [1380–1410], Int7 (phenols) = [1410–1450], Int8 (protein amide II) = [1490–1580], Int9 (protein β-loop) = [1600–1650], Int10 (aromatic compounds) = [1680–1720], and Int11 (esterified carboxylic groups of pectin) = [1720–1760] [41].

Total sugar determination (soluble and insoluble—cellulose—): The total sugar concentration in the CW extract was determined using the phenol-sulfuric acid method [80]. A 400 μL aqueous sample was mixed with 400 μL of 5% phenol in a glass tube, followed by the addition of 2 mL concentrated H_2_SO_4_, with vigorous shaking. After cooling to room temperature for 5–10 min, absorbance was measured at 490 nm (SPECTROstar Nano BMG Labtech, Ortenberg, Germany) using a glucose standard curve as a reference.

Uronic acid determination: Uronic acids were determined colorimetrically using an adapted 3-hydroxybiphenyl method [81]. A 400 µL aqueous sample was mixed with 2.4 mL of 12.5 mM sodium tetraborate in concentrated H_2_SO_4_, chilled in an ice bath, and homogenized. The mixture was boiled for 5 min and re-chilled, and then 40 µL of 0.15% 3-hydroxydiphenylphenol (*w:v*; in 0.5% NaOH) was added and vigorously mixed. After 10 min, the absorbance was measured at 520 nm. A standard curve with galacturonic acid (Sigma Aldrich, Darmstadt, Germany) served as the reference.

#### 4.3.2. Nutraceutical Analysis

Nutraceutical compound (NUTC) extraction: Aqueous or methanolic extracts (40 mg/mL) of fresh onion adaxial epidermal material were prepared for phenol, flavonoid, and antioxidant activity quantification. The material was homogenized with an automatic mill in liquid nitrogen and stored at −80 °C. After adding distilled water, the mixture was vigorously homogenized and centrifuged (10 min; 13,000× *g*; 4 °C). The supernatant was transferred to a new 1.5 mL tube and stored at −20 °C until analysis.

Antioxidant activity quantification: Antioxidant activity was quantified using a modified DPPH method with a Trolox standard curve [82]. Methanol extract (50 µL) was mixed with 250 µL of 2,2-diphenyl-1-picrylhydrazyl (DPPH) solution (20 mg/L) and incubated in the dark for 30 min. A Trolox standard curve (Trolox^®^ 97%, ACROS Organics, Geel, Belgium) was prepared in parallel. Absorbance was measured at 517 nm, and antioxidant activity was expressed in µmol Trolox equivalents (TE) per 100 g of sample fresh weight.

Phenol quantification: Total phenols were quantified using a modified Folin–Ciocalteu method [82]. Aqueous extract (250 µL) was mixed with 12.5 µL of Folin–Ciocalteu reagent (Merck, Darmstadt, Germany) and incubated in the dark for 15 min. Next, 37.5 µL of 20% Na_2_CO_3_ (*w/v*) was added, and the mixture was kept in the dark for 2 h. A gallic acid standard curve was prepared in parallel, and absorbance was measured at 765 nm after incubation.

Flavonoid quantification: Total flavonoids were quantified using a method modified from [82]. Methanol extract (166 µL) was mixed with 10 µL of 5% NaNO_2_ and allowed to react for 5 min. Then, 10 µL of 10% Al(NO_3_)_3_ was added and incubated for another 5 min, followed by 66 µL of 1 M NaOH and 80 µL of distilled water. Absorbance was measured at 510 nm, using a quercetin standard curve as a reference.

#### 4.3.3. Total Protein Quantification

Total protein extraction from adaxial epidermal cell wall: Proteins were extracted from previously homogenized material stored at −20 °C using a modified method from [67]. Samples (3 mL/g fresh weight) were homogenized in extraction buffer (0.5 g/mL) containing 40 mM sodium acetate buffer (pH 5.0), 13 mM CDTA, 10 mM β-mercaptoethanol, 1% PVP 10, and 1 M NaCl. The mixture was agitated overnight at 4 °C and then centrifuged for 15 min. The residue was dissolved in 50 mM sodium acetate buffer (pH 6.0) for quantification, and the supernatant was transferred to a new 1.5 mL tube and stored at −20 °C for further analysis.

Protein quantification: Protein content was determined using the Bradford method [83] with NZYBradford reagent (Nzytech, Lisbon, Portugal), following the manufacturer’s instructions. Bradford reagent (590 µL) was added to 10 µL of sample, mixed, and incubated for 10 min before measuring absorbance at 595 nm. Bovine serum albumin (BSA; Pierce™, Thermo Fisher Scientific, Rockford, IL, USA) was used as the standard.

### 4.4. CWE Gene Expression

#### 4.4.1. CWE Gene Selection

To evaluate the expression level of genes encoding CWEs, an in silico study was carried out. Genes encoding CWEs from *Allium cepa* L. were identified from the NCBI GenBank database (Release 245.0, 15 August 2021; accessed on September 2021). For each gene, sequence homology was analyzed through NCBI BLAST (https://blast.ncbi.nlm.nih.gov/Blast.cgi (accessed on 19 March 2025)), comparing the *Allium cepa* L. gene sequences to corresponding sequences from related species and genera within the same family. The most conserved sequences were selected to ensure relevance and specificity for *Allium cepa* L. Primer design for these genes was carried out using the PrimerQuest tool (IDT, Integrated DNA Technologies, Leuven, Belgium), with selection criteria emphasizing a GC content of approximately 50%, a melting temperature (Tm) around 60 °C, primer lengths between 18 and 30 bases, and amplicon sizes between 80 and 200 base pairs. Each primer was further evaluated for secondary structure formation, such as hairpin loops, to ensure efficient binding. Finally, primer specificity was confirmed by running each sequence through NCBI BLAST to avoid off-target amplification of other genes. A total of 10 genes (expansin, fucosidase, galactosidase, glucanase, PME, PLL, PG, xylanase, XTH, and xylosidase) encoding CWEs participating in the regulation and maintenance of the CW were selected. A housekeeping gene (actin) was also chosen to allow gene expression to be normalized in the RT-qPCR assays.

#### 4.4.2. RNA Extraction

For RNA extraction, the adaxial epidermis of layers S1, S2, and S6 was collected in 1.5 mL tubes and immediately frozen in liquid nitrogen. The frozen samples were ground with an automatic mill (Ventura mix; Madespa, Toledo, Spain). Total RNA was extracted using an RNA extraction kit (NZY Total RNA Isolation kit, Nzytech) and diluted in RNase-free water. RNA samples were quantified using a SPECTROstar Nano droplet reader (BMG Labtech).

#### 4.4.3. Gene Expression Analysis Using RT-qPCR

After the efficiency of all the oligonucleotides (Integrated DNA Technologies, Inc. IDT) used in this analysis was checked (slope > 3), real-time quantitative PCR amplification (RT-qPCR) and detection were carried out on a MyGo Pro System (KromoGEN, Nilüfer-Bursa, Turkey) using a RT-qPCR One-Step kit (One-step NZYSpeedy RT-qPCR Green kit, Nzytech) following the protocol’s instructions. The PCR conditions were as follows: 50 °C for 20 min, then 95 °C for 5 min, and then 50 cycles of 95 °C for 10 s to 60 °C for 30 s. At the end of each reaction, the melting curve of each gene was evaluated by programming a rise from 60 °C to 97 °C at a speed of 0.1 °C/s to ensure that only single products were formed. The products were verified electrophoretically on a 1% agarose gel. The expression levels of each gene, relative to the housekeeping gene actin (GeneBank locus GU570135.2, https://www.ncbi.nlm.nih.gov/nuccore/GU570135.2 (accessed on 18 March 2025)), were determined using the Pfaffl method [84]. All the oligonucleotides used for RT-qPCR are summarized in Appendix A.

### 4.5. Supervised Analysis and Predictive Models

The database was composed of five datasets (https://github.com/JJ-Lab/onion_epidermis (accessed on 18 March 2025)): (1) cellular morphology dataset (cell size and cell number; S1–S6 layers; B, M, and U zones); (2) FTIR dataset (absorbance; S1, S2, and S6); (3) CWC dataset (total sugars, uronic acids (pectins), total insoluble sugars (α-cellulose), and reducing sugars; S1, S2, and S6); (4) NUTC dataset (total antioxidants, total phenols, flavonoids, and total proteins; S1, S2, and S6); and (5) gene expression dataset (enzymes expansin, fucosidase, galactosidase, glucanase, PME, PLL, PG, xylanase, XTH and xylosidase; S1, S2, and S6).

Using the data obtained from the analysis of plant material, we designed different predictive models. As the data were supervised, and we wanted to determine which discrete layer they belong to, so all the models we made were classification models. Based on the properties analyzed, we achieved better results with different algorithms. We used the vast majority of standard algorithms for these models: Logistic Regressions, Support Vector Machines (SVM), k-Nearest Neighbors (kNN), Decision Tree, Random Forest, Gradient Boosting, and Neural Networks. The training–test split was 70%–30%, which is standard. To score the models we used classification accuracy and precision tools. Classification accuracy (CA) measures the proportion of correctly classified cases out of the total number of cases. Thus, CA = (TP + TN)/(TP + TN + FP + FN), where: TP (true positives) = correctly classified positive cases; TN (true negatives) = correctly classified negative cases; FP (false positives) = negative cases incorrectly classified as positive; and FN (false negatives) = positive cases incorrectly classified as negative. Precision measures how many of the predicted positive cases are actually correct. Precision = TP/(TP + FP). High accuracy means fewer false positives (FPs), indicating that the model is more likely to be correct when it predicts a positive case. We also reviewed the confusion matrix to check that the models predicted the three layers in a similar way, with no layer having a very low prediction. To build these models, we used the Orange3 tool [85], programmed in Python 3.8.5 (https://www.python.org/), and applied Python scripts.

#### 4.5.1. FTIR Analysis

The 11 FTIR spectral intervals demonstrating significant inter-layer differences were selected for detailed characterization. The integral of the curve of the *m/z* spectra within these intervals was calculated, providing a quantitative measure of the cumulative differences among layers. These FTIR data integrals were subsequently employed as features for supervised classification algorithms. A variety of ML algorithms was used to predict the specific layer (target variable) from the FTIR data integrals (input variables), and confusion matrices were generated to verify good prediction in all layers. These matrices provided a comprehensive representation of the percentage of correctly and incorrectly classified layers.

#### 4.5.2. CWC and NUTC Analysis

To classify the onion layers (S1, S2, and S6) depending on CW and nutraceutical composition, we applied six different supervised ML algorithms. The table in Figure 5 indicates the methods that provided the best results in each case.

#### 4.5.3. Correlational Analysis

Correlations between different CWC-NUTC features and gene expression were performed with unpaired and unbalanced data. For this reason, we used a bootstrap method [86] to determine an average Pearson correlation coefficient. Firstly, we normalized gene expression data within the same onion. For each datapoint, we subtracted the mean and divided by the standard deviation. Secondly, each sample datapoint was composed of one value for layer 1, one for layer 2, and one for layer 6. As the samples used in the chemical analysis and gene expression analysis corresponded to different onions, we randomly selected one sample (with the values of the three layers) from the biochemical data and another random sample (also with the values of the three layers) from the gene expression data; matching the corresponding values of biochemistry and gene expression of the same layer, we analyzed the linear correlation of each bootstrap sample. The basic idea of any bootstrap method is to repeat many random resamplings (with replacement) from the original data to generate a distribution of estimates. Plotting these pairs of values, we tried to perform least squares fit regression for every resampling. Pearson correlation can be performed for the sampled data pairs (bootstrap sample), and a correlation coefficient distribution can be obtained. This distribution can be used for hypothesis testing, standard error estimation, and confidence interval calculation.

## 5. Conclusions

This study lays a robust foundation for elucidating the genomic, molecular, structural, and biochemical mechanisms underlying onion epidermis growth. By integrating spectrophotometric quantifications, profiling FTIR spectroscopy, enzymatic gene expression, and ML modeling, it provides cutting-edge insights into the dynamics of CWCs and NUTCs, underscoring their essential roles in plant development and stress response. These advanced methodologies deliver unparalleled precision in linking molecular and biochemical traits to specific growth stages, paving the way for deeper exploration of developmental regulation.

Future studies should build on this framework by incorporating multi-omics datasets to enhance predictive accuracy and uncover the intricate regulatory networks that govern CW dynamics. Moreover, examining the functional roles of CWEs across diverse environmental scenarios will further illuminate the mechanisms underlying plant resilience and adaptability. These findings present transformative potential for agricultural innovation, especially in breeding high-yield, stress-resilient crop varieties adapted to a wide range of climatic conditions. By seamlessly integrating fundamental research with practical applications, this work significantly advances the pursuit of sustainable agricultural practices and global food security.

## Figures and Tables

**Figure 1 ijms-26-02946-f001:**
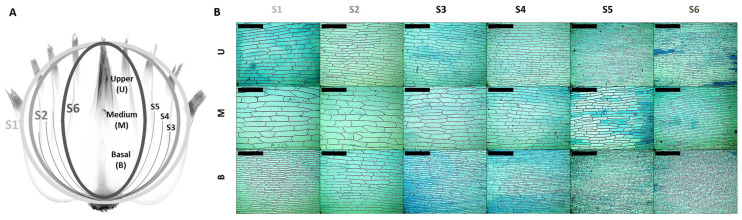
Structural organization and epidermal cell morphology in onion bulb layers. (**A**) Schema representing onion layers (S1 to S6) and the basal (B), medium (M), and upper (U) zones; (**B**) trypan blue staining of onion epidermal cells in layers S1–S6 and the B, M, and U zones. Layers S1, S2, and S6 are represented as three shades of grey. Scale bar: 400 μm.

**Figure 2 ijms-26-02946-f002:**
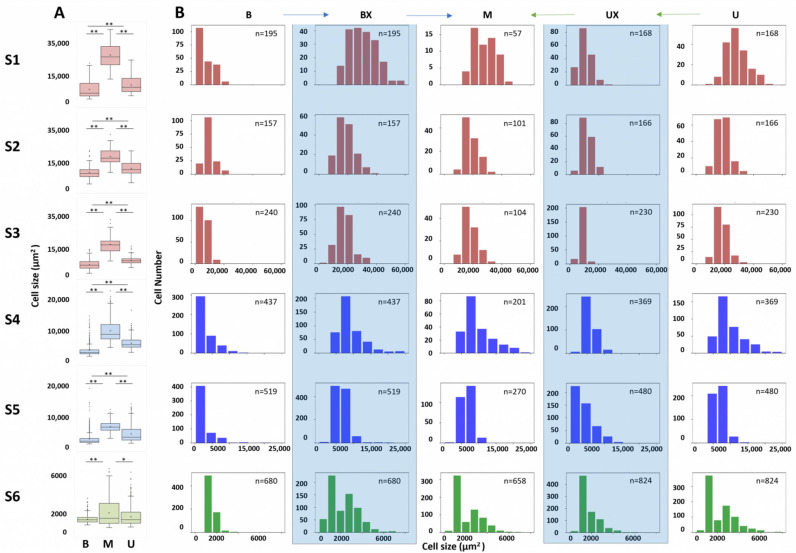
Distribution of cell sizes and numbers in the different areas (basal, middle, and upper) of the onion layers (S1–S6). (**A**) Cell sizes in the different areas (basal, middle, and upper) of the onion layers (S1–S6). Kruskal–Wallis followed by Dunn’s test was used to analyze data from three independent biological repeats. Asterisks indicate significant differences * (*p* ≤ 0.001); ** (*p* ≤ 0.0001). (**B**) Distribution of cell sizes and numbers in the different areas (basal, middle, and upper) of the onion layers (S1–S6). The histograms show three colors associated with three different scales: S1, S2, and S3 contained full-scale cells (in red), S4 and S5 contained cells on an intermediate scale (in blue), and S6 exhibited smaller scale cells (in green). The columns with a light blue background correspond to the modified histograms. Graphical verification of the hypothesis of proportional cell size from the basal area (B and BX) to the middle area (M), with blue arrows and proportional cell size from the upper area (U and UX) to the middle, with green arrows.

**Figure 3 ijms-26-02946-f003:**
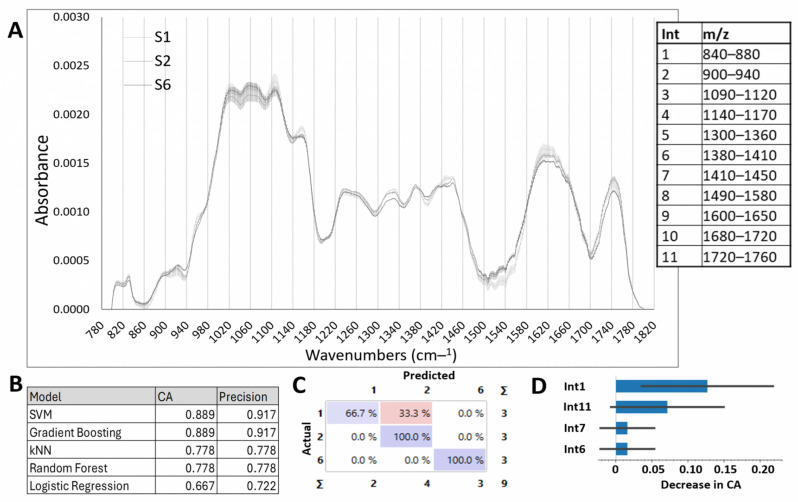
Cell wall composition of the epidermal cells of onion layers and classification model. (**A**) FTIR analysis of the cell wall in layers S1, S2, and S6, showing eleven significant difference intervals. Spectra show the mean value (n = 3). (**B**) Classification accuracy (CA) and precision of the different models used in the eleven ranges from the FTIR dataset. (**C**) Confusion matrix for the SVM model (FTIR dataset). The confusion matrix shows the percentage of correctly (in purple) and incorrectly (in pink) classified instances. Columns represent predicted values, while rows represent true values. (**D**) Feature importance for the SVM model applied to the eleven ranges from the FTIR dataset.

**Figure 4 ijms-26-02946-f004:**
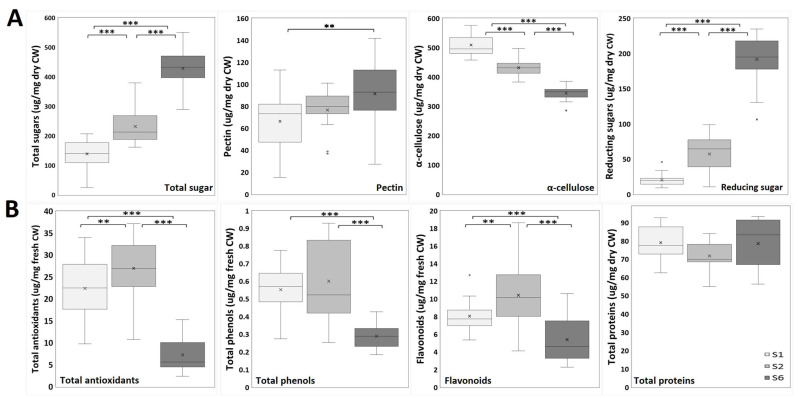
Spectroscopic analysis of onion epidermal cells. (**A**) Onion epidermis cell wall analysis (n = 18): total sugars; uronic acids (pectins); total insoluble sugars (α-cellulose); reducing sugars. (**B**) Onion epidermis nutraceutical analysis (n = 18–33): total antioxidants; total phenols; flavonoids; total proteins. Unpaired Student’s *t*-test was used to analyze data from three independent biological repeats. Asterisks indicate significant differences ** (*p* ≤ 0.01); *** (*p* ≤ 0.001).

**Figure 5 ijms-26-02946-f005:**
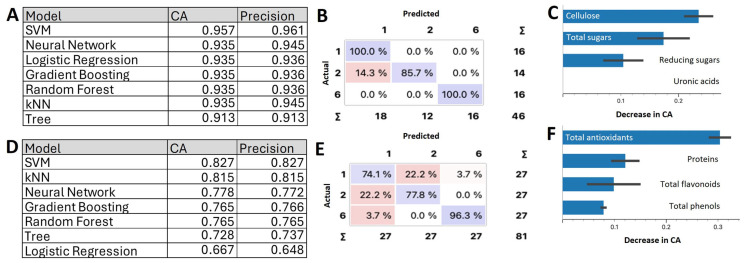
Scale classification by machine learning. (**A**) Classification accuracy (CA) and precision for the different models used in the CWC dataset. (**B**) Confusion matrix for the SVM model (CW biochemical dataset). The confusion matrix shows the percentage of correctly (in purple) and incorrectly (in pink) classified instances. Columns represent predicted values, while rows represent true values. (**C**) Feature importance for the SVM model. The histogram shows the significance of features importance for the SVM model in the classification of the CWC dataset. (**D**) Classification accuracy and precision for the different models used in the NUTC dataset. The tables indicate the methods that provided the best result in each case. (**E**) Confusion matrix for the SVM model (nutraceutical dataset). The matrix highlights well-predicted values along the diagonal, with S6 again being the best-predicted category. (**F**) Feature importance for the SVM model applied to the NUTC dataset.

**Figure 6 ijms-26-02946-f006:**
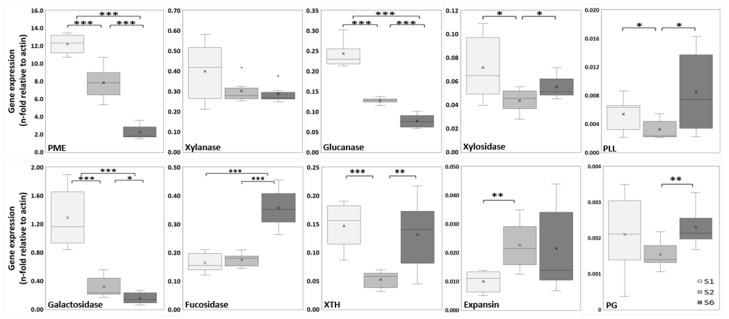
Gene expression of the cell wall enzymes pectin methylesterase (PME), xylanase, glucanase, xylosidase, pectin/pectate lyase-like (PLL), galactosidase, fucosidase, xyloglucan endotransglycosylase/hydrolase (XTH), expansin and polygalacturonase (PG) and, as determined by RT-qPCR. The expression of CWE genes through the three growth stages was normalized to onion actin expression. Data from three independent biological repeats were analyzed by unpaired Student’s *t*-test (n = 6). Asterisks indicate significant differences * (*p* ≤ 0.05); ** (*p* ≤ 0.01); *** (*p* ≤ 0.001).

**Table 1 ijms-26-02946-t001:** Pearson coefficients for the different pairs of variables. The blue text shows coefficients above +0.800, and the red text shows coefficients below −0.800.

Pearson Coefficient	Total Sugar	Uronic Acid	α-Cellulose	Reducing Sugar	Total Antioxidants	Total Phenols	Flavonoids	Total Protein
XTH	0.0740	0.0543	0.0793	0.1834	−0.3271	−0.2462	−0.3853	0.2013
Glucanase	0.7180	0.0660	−0.5599	0.7575	−0.8687	−0.7716	−0.7780	0.3506
Xylanase	−0.4785	−0.0111	0.4158	−0.4027	0.2773	0.2604	0.1591	−0.0376
Expansin	0.3501	−0.1167	−0.4240	0.2851	−0.0803	−0.0824	0.0091	−0.0138
Galactosidase	−0.7815	−0.0727	0.7560	−0.7537	0.5477	0.6119	0.3442	−0.0613
Fucosidase	0.9577	0.1292	−0.8385	0.9662	−0.9134	−0.8780	−0.6842	0.1053
PG	0.2823	0.0370	−0.2317	0.3200	−0.5384	−0.4106	−0.4796	0.2992
PLL18	0.5125	−0.0807	−0.4253	0.6115	−0.6310	−0.5103	−0.5362	0.2233
PME	−0.9467	0.0734	0.8774	−0.9482	0.7665	0.8163	0.5503	−0.1846
Xylosidase	−0.1705	0.0094	0.1314	−0.0698	−0.1008	−0.1306	−0.3869	0.1985

## Data Availability

Data are contained within the article and Appendix A.

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
