# Peer review of "Cell Wall–Based Machine Learning Models to Predict Plant Growth Using Onion Epidermis"

_ijms, 2025, doi:10.3390/ijms26072946_

Round 1
Reviewer 1 Report
Comments and Suggestions for Authors
The study looks at using machine learning to predict the growth stages in onion epidermis, leveraging FTIR spectroscopy, biochemical analysis, and cell wall composition. While it offers some useful insights into plant growth, the innovations aren't as groundbreaking when compared to similar studies. Here's a more casual breakdown of the key points:
The study mainly relies on traditional machine learning models like SVM, kNN, and neural networks for classifying growth stages. But these methods might not capture all the complex patterns in the data, especially when compared to more advanced techniques like deep learning or ensemble methods. Deep learning models, for example, can offer more accurate predictions and reveal hidden patterns in biological data (Jeong, Seungtaek, et al. "Incorporation of machine learning and deep neural network approaches into a remote sensing-integrated crop model for the simulation of rice growth." Scientific reports 12.1 (2022): 9030.). Ensemble methods, such as random forests and gradient boosting, could also improve the accuracy and stability of the models but aren't explored here.
While the study looks at FTIR data, biochemical parameters, and enzyme expression, it doesn't integrate other omics approaches (like genomic, proteomic, and metabolomic data). These kinds of multi-omics approaches are now standard in plant biology and could greatly strengthen the predictive models (van Dijk, Aalt Dirk Jan, et al. "Machine learning in plant science and plant breeding." Iscience 24.1 (2021).).
The research focuses on onion epidermis and its growth linked to cell wall composition. However, applying these findings to other plant species or different parts of the onion could make the research more broadly applicable. Other studies have used machine learning to model plant growth across different species and environments, offering more general insights (Araújo, Sara Oleiro, et al. "Machine learning applications in agriculture: current trends, challenges, and future perspectives." Agronomy 13.12 (2023): 2976.).
The study mainly looks at the physical properties of the cell wall and how they affect growth. But there's recent research suggesting that protein-peptide-polysaccharide interactions play a significant role in cell wall mechanics. This hasn't been touched on in the study, even though it could be important for understanding how the cell wall remodels and impacts plant growth (Civantos-Gómez, Iciar, et al. "Climate change conditions the selection of rust-resistant candidate wild lentil populations for in situ conservation." Frontiers in Plant Science 13 (2022): 1010799.).
The study looks at ten cell wall enzyme genes but doesn't fully explore their functions at different growth stages. There are many less-studied enzymes in plant cell wall metabolism that could offer deeper insights into growth regulation. Research in this area could help us better understand enzyme roles in plant resilience and growth (Yan, Jun, and Xiangfeng Wang. "Machine learning bridges omics sciences and plant breeding." Trends in Plant Science 28.2 (2023): 199-210.).
To sum it up, while the study does a good job of laying the groundwork for understanding plant cell wall dynamics, there's definitely room to enhance the research by using more advanced machine learning techniques, integrating multi-omics data, and exploring more biochemical factors. Expanding the study's scope and diving into new technologies and biochemical components could make the research even more comprehensive and applicable.
Author Response
For research article
|
Response to Reviewer 1 Comments
|
||
|
1. Summary |
|
|
|
Thank you very much for taking the time to review this manuscript. Please find the detailed responses below and the corresponding revisions/corrections highlighted/in track changes in the re-submitted files.
|
||
|
2. Questions for General Evaluation |
Reviewer’s Evaluation |
Response and Revisions |
|
Does the introduction provide sufficient background and include all relevant references? |
Yes |
See point-by-point response to comments and an suggestions for authors. |
|
Is the research design appropriate? |
Yes |
|
|
Are the methods adequately described? |
Yes |
|
|
Are the results clearly presented? |
Yes |
|
|
Are the conclusions supported by the results?
|
Yes |
|
|
3. Point-by-point response to Comments and Suggestions for Authors |
||
|
Comments 1: The study mainly relies on traditional machine learning models like SVM, kNN, and neural networks for classifying growth stages. But these methods might not capture all the complex patterns in the data, especially when compared to more advanced techniques like deep learning or ensemble methods. Deep learning models, for example, can offer more accurate predictions and reveal hidden patterns in biological data (Jeong, Seungtaek, et al. "Incorporation of machine learning and deep neural network approaches into a remote sensing-integrated crop model for the simulation of rice growth." Scientific reports 12.1 (2022): 9030.). Ensemble methods, such as random forests and gradient boosting, could also improve the accuracy and stability of the models but aren't explored here.
|
||
|
Response 1: Thank you for pointing this out, we agree that it deserves clarification. We understand this comment about the choice of machine learning models. However, it has been shown that, with our data, simple machine learning models provide excellent results, making the use of more complex models unnecessary for this study. Thus, Deep learning models will hardly improve the accuracy of the biological predictions obtained in this article with simple machine learning methods. Furthermore, our data are not suitable and insufficient for the application of Deep learning models. On the other hand, simpler models are more scientifically applicable and more transferable to society, reinforcing the practical value of our chosen approach. Regarding the ensemble methods, we thank the reviewer for not making it sufficiently clear and we point out to the reviewer that these methods, such as random forest and gradient boosting, have been used in this article (Figure 3B, Figure 5A and 5D) and as indicated in section 4.5 of Materials and Methods, lines 680-681. We also appreciate the reviewer’s reference to Jeong et al. (2022). However, we would like to note that this article is already cited in our manuscript as reference [35]. Specifically, we acknowledge its contribution in the context of genotype-phenotype mapping and plant genotype classification under diverse conditions (Lines 153–155). We do not consider it necessary to change the text of the manuscript.
|
||
|
Comments 2: While the study looks at FTIR data, biochemical parameters, and enzyme expression, it doesn't integrate other omics approaches (like genomic, proteomic, and metabolomic data). These kinds of multi-omics approaches are now standard in plant biology and could greatly strengthen the predictive models (van Dijk, Aalt Dirk Jan, et al. "Machine learning in plant science and plant breeding." Iscience 24.1 (2021).
Response 2: We understand and appreciate the reviewer's comment regarding the integration of multi-omics approaches. However, one of the objectives of this work is to develop models that are practical and easily applicable in real-world agricultural and breeding programs. To do so, it is necessary to apply accessible and economic analyses that can be applied by plant breeding companies, such as the biochemical parameters analyzed in our work. As rightly indicated by the referee, and as discussed in our manuscript (REF. [34]), omics analyses provide a lot of information and allow obtaining strengthened predictive models, however, their high cost and difficulty in interpreting the information obtained in many cases makes them unviable for breeding companies. Consequently, we do not believe it is necessary to make any changes to the manuscript to emphasize this point.
|
||
|
Comments 3: The research focuses on onion epidermis and its growth linked to cell wall composition. However, applying these findings to other plant species or different parts of the onion could make the research more broadly applicable. Other studies have used machine learning to model plant growth across different species and environments, offering more general insights (Araújo, Sara Oleiro, et al. "Machine learning applications in agriculture: current trends, challenges, and future perspectives." Agronomy 13.12 (2023): 2976.).
|
||
|
Response 3: Thank you for pointing this out. Indeed, as indicated by the referee and the article already cited in the text (Ref. [32]), we recognize the importance of applying ML models across different species and environments. While our current work establishes a focused and interpretable model for onion epidermis growth (as a single individual synchronous model), we are actively working on extending this approach to other plant tissues and species. In fact, in an upcoming article, we will investigate asynchronous growth across multiple individuals to broaden the applicability of our findings. We do not consider it necessary to change the text in the manuscript.
|
||
|
Comments 4: The study mainly looks at the physical properties of the cell wall and how they affect growth. But there's recent research suggesting that protein-peptide-polysaccharide interactions play a significant role in cell wall mechanics. This hasn't been touched on in the study, even though it could be important for understanding how the cell wall remodels and impacts plant growth (Civantos-Gómez, Iciar, et al. "Climate change conditions the selection of rust-resistant candidate wild lentil populations for in situ conservation." Frontiers in Plant Science 13 (2022): 1010799.).
Response 4: We agree with the referee's comment. While our study does not molecularly analyze protein-peptide-polysaccharide interactions, it does investigate the physical properties of the cell wall (CW), which are ultimately shaped by the enzymatic activity of CW enzymes (CWEs) on polysaccharides. These interactions play a crucial role in governing plant growth. The indicated article is already cited in the text (Ref. [33]), but does not refer to the previously indicated topic. We do not consider it necessary to change the text in the manuscript.
|
||
|
Comments 5: The study looks at ten cell wall enzyme genes but doesn't fully explore their functions at different growth stages. There are many less-studied enzymes in plant cell wall metabolism that could offer deeper insights into growth regulation. Research in this area could help us better understand enzyme roles in plant resilience and growth (Yan, Jun, and Xiangfeng Wang. "Machine learning bridges omics sciences and plant breeding." Trends in Plant Science 28.2 (2023): 199-210.).
|
||
|
Response 5: Thank you for this observation. We recognize that less studied CW enzymes may also play important roles in regulating plant growth and resilience, as highlighted in Yan and Wang (2023). However, this study discusses how machine learning can bridge fundamental plant research and applied breeding by integrating multi-omics data. It further explores the applications of machine learning in gene regulation, gene discovery, and trait prediction, emphasizing how data-driven approaches can improve our understanding of key genes involved in plant development. We will incorporate this perspective in our revised manuscript. It is not usual for publications studying CWE to address the study of 10 different genes, but rather they usually analyze one of the genes and/or several isozymes of the same multigene family, consequently studying the modification of a single cell wall component. Thus, in our work we studied the different layers of the onion to analyse their differential expression throughout plant growth, from the youngest and inner tissues (S6) to the oldest and outer ones (S1). In this way, our work helps us to better understand the role of the 10 CWEs in plant growth. |
||
|
We do not consider it necessary to change the text in the manuscript. |
||
|
|
||
|
4. Response to Comments on the Quality of English Language |
||
|
Point 1: The English is fine and does not require any improvement. |
||
|
Response 1: No changes are included.
|
||
|
5. Additional clarifications |
||
|
Again, thank you very much for taking the time to review this manuscript and propose changes that improve our publication. |
||

Reviewer 2 Report
Comments and Suggestions for Authors
The authors developed machine learning models to predict the growth stages of onion epidermal cells by integrating FTIR data, biochemical parameters, and cell wall enzyme gene expression profiles. I have the following suggestions:
- Please consider reorganizing the structure of this paper to follow this order: (1) Introduction; (2) Data and method; (3) Result; (4) Discussion; and (5) Conclusion.
- In Figure 3B, Figure 5A and 5D, and Table 1, I assume that all commas “,” in the tables should be replaced with decimal points “.”.
- Figure3A: The absorbance of S1, S2, and S6 are very similar (e.g., in Int1 and Int11), while Inl1 was the most important feature in the machine learning model. Please explain how these similar features can help distinguish different growing stages.
- Page 674-685: When developing ML models, we usually divide the dataset into a training set (used to train the models) and a validation set (not seen during training). Please specify the number of data points used for training and validation, respectively.
- Page 674-685: Please also clarify how classification accuracy and precision were calculated.
Author Response
For research article
|
Response to Reviewer 2 Comments
|
||
|
1. Summary |
|
|
|
Thank you very much for taking the time to review this manuscript. Please find the detailed responses below and the corresponding revisions/corrections highlighted/in track changes in the re-submitted files.
|
||
|
2. Questions for General Evaluation |
Reviewer’s Evaluation |
Response and Revisions |
|
Does the introduction provide sufficient background and include all relevant references? |
Yes |
See point-by-point response to comments and suggestions for authors. |
|
Is the research design appropriate? |
Yes |
|
|
Are the methods adequately described? |
Must be improved |
|
|
Are the results clearly presented? |
Can be improved |
|
|
Are the conclusions supported by the results? |
Yes |
|
|
3. Point-by-point response to Comments and Suggestions for Authors
|
||
|
Comments 1: Please consider reorganizing the structure of this paper to follow this order: (1) Introduction; (2) Data and method; (3) Result; (4) Discussion; and (5) Conclusion.
|
||
|
Response 1: Thank you for pointing this out. We understand and share the reviewer's point of view. However, the IJMS template specifies the order of the parts of the article that are included and have been followed in the manuscript. We do not consider it necessary to change the text in the manuscript.
|
||
|
Comments 2: Figure 3B, Figure 5A and 5D, and Table 1, I assume that all commas “,” in the tables should be replaced with decimal points “.”.
|
||
|
Response 2: Agree. We have, accordingly, changed the "," by "." to correct this point. It is changed in the indicated figures 3B, 5A, 5D, table 1 and in supplementary material table S2 of the manuscript.
|
||
|
Comments 3: Figure3A: The absorbance of S1, S2, and S6 are very similar (e.g., in Int1 and Int11), while Inl1 was the most important feature in the machine learning model. Please explain how these similar features can help distinguish different growing stages.
|
||
|
Response 3: We appreciate this comment because it may be a source of confusion for other readers. Regarding the question about Figure 3, we would like to point out that Figure 3B shows a table with the 5 ordered models that offered the best classification data Accuracy (CA) and Precision in the eleven ranges FTIR dataset. It should be noted that the first two models (SVM and Gradient Boosting) offered the same results for CA and precision. Figure 3C shows the Confusion Matrix for one of the two best models obtained (SVM). Figure 3D shows the Feature importance for the SVM model applied to the eleven ranges FTIR dataset. Indeed, if we show the graph of Feature importance for the Gradient Boosting model applied to the eleven ranges FTIR dataset (attached below), Int8 appears as an important feature. In this case, the confusion matrix obtained is similar to the one presented in Figure 3C for the SVM model (100% positive assignment in S1 and S6, and 66% in S2). In summary, depending on the model you choose, the internal statistical algorithm will use some features or others to classify. The classification prediction is not altered by this, which is supported by the same CA and precision values, i.e. the final result is not modified. In our case, we have chosen the SVM model because the features obtained allow us to better illustrate the rest of the results and support the discussion.
We consider it necessary to add to the text of the manuscript: line 287: SVM line 289: to the SVM model line 427: to the SVM model
|
||
|
Comments 4: Page 674-685: When developing ML models, we usually divide the dataset into a training set (used to train the models) and a validation set (not seen during training). Please specify the number of data points used for training and validation, respectively.
|
||
|
Response 4: Thank you for highlighting this aspect. The training-test split was 70%-30%, which is usually the standard. It is changed in the material and methods section 4.5 of the manuscript (lines 682-683).
|
||
|
|
||
|
Comments 5: Page 674-685: Please also clarify how classification accuracy and precision were calculated.
|
||
|
Response 5: This is an important point, and we are happy to clarify it. Classification accuracy (CA) measures the proportion of correctly classified cases out of the total number of cases. Thus, CA = (TP + TN) / (TP + TN + FP + FN), where: TP (True Positives) = Correctly classified positive cases; TN (True Negatives) = Correctly classified negative cases; FP (False Positives) = Negative cases incorrectly classified as positive and FN (False Negatives) = Positive cases incorrectly classified as negative. Precision measures how many of the predicted positive cases are actually correct. Precision = TP / (TP + FP). High accuracy means fewer false positives (FP), indicating that it is more likely to be correct when the model predicts a positive case. It is changed in the material and methods sections 4.5 of the manuscript (lines 684-691). |
||
|
|
||
|
|
||
|
4. Response to Comments on the Quality of English Language |
|
Point 1: The English is fine and does not require any improvement. |
|
Response 1: No changes are included.
|
|
5. Additional clarifications |
|
Again, thank you very much for taking the time to review this manuscript and propose changes that improve our publication. |

Reviewer 3 Report
Comments and Suggestions for Authors
- Figure S2 - completely unreadable. Correct it.
- For Figure 2 the nomenclature in Figure S1-S6 and in the description can be confusing as the authors use similarly for supplemenatary Figure S2. Correct it.
- Figure 3 should be corrected - the decimal separator should be a dot. This applies to axes and tables. Correct it.
- The article uses different words related to the evaluation of metrics for models. This repeatedly turns on whether we are dealing explicitly with classification or regression. The development of machine learning methods translates into the selection of algorithms and loss functions. In view of this, substantively the entire article should be improved.
- When evaluating metrics from a classification point of view, consider precision in addition to recall. Correct it.
- Line 555: use the dimensional tolerance for values this figure unacceptable. This also applies to any measurement taken in the research. Correct it.
- As part of the data on the design of machine learning models, very little has been described by the authors. Libraries are required, as well as relevant literature from which practical knowledge has been derived.
- No key action what did the dataset look like for this problem - how many cases, what input variables, what output variables were included in the collection?
- What did the test and training set look like?
- What proportions were used between the test and training set?
- Was data normalisation applied?
- In conclusion, the topic should necessarily be improved. It is not good for the research question posed.
- The topic is not very much about machine learning I have concerns about the use of these methods in this article.
- The confusion matrix should specify on which set the results are interpreted.
- Dataset is missing. Dataset should be posted e.g. Mendeley Data or similar repository for data access.
Comments on the Quality of English LanguageThe English could be improved to more clearly express the research.
Author Response
For research article
|
|
Response to Reviewer 3 Comments
|
||
|
|
1. Summary |
|
|
|
|
Thank you very much for taking the time to review this manuscript. Please find the detailed responses below and the corresponding revisions/corrections highlighted/in track changes in the re-submitted files.
|
||
|
|
2. Questions for General Evaluation |
Reviewer’s Evaluation |
Response and Revisions |
|
|
Does the introduction provide sufficient background and include all relevant references? |
Yes |
See point-by-point response to comments and a suggestions for authors. |
|
|
Is the research design appropriate? |
Can be improved |
|
|
|
Are the methods adequately described? |
Must be improved |
|
|
|
Are the results clearly presented? |
Must be improved |
|
|
|
Are the conclusions supported by the results? |
Can be improved |
|
|
|
3. Point-by-point response to Comments and Suggestions for Authors |
||
|
|
Comments 1: Figure S2 - completely unreadable. Correct it.
|
||
|
|
Response 1: Thank you for your comment. Figure S2 has been split into two panels to improve readability, and the figure legend has been modified to also address Comment 2.
The changes have been made in Figure S2 and the figure caption in the supplementary document to the manuscript.
|
||
|
|
|
||
|
|
Comments 2: For Figure 2 the nomenclature in Figure S1-S6 and in the description can be confusing as the authors use similarly for supplementary Figure S2.
|
||
|
|
Response 2: Thank you for pointing this out. Figure 2 shows the distribution of cell sizes and numbers in the different areas (basal, middle and top) of the 6 onion layers (S1-S6) studied. Panel (A) shows the cell sizes in the different areas (basal, middle and top) of the onion layers (S1-S6) and panel (B) shows the distribution of cell sizes and numbers in the different areas (basal, middle and top) of the onion layers (S1-S6). However, Figure S2 shows the correlations between 4 CWC (A) and 4 NUTC (B) traits and the expression of 10 CWE genes using bootstrap methods only in the middle part of S1 (circles), S2 (squares) and S6 (crosses). It changed the figure legend of the Figure S2 in supplementary document to the manuscript.
|
||
|
|
Comments 3: Figure 3 should be corrected - the decimal separator should be a dot. This applies to axes and tables. |
||
|
|
Response 3: Agree. We have, accordingly, changed the "," by "." to correct this point. It is changed in the figures 3B, 5A, 5D, table 1 and in supplementary material table S2 of the manuscript.
|
||
|
|
Comments 4: The article uses different words related to the evaluation of metrics for models. This repeatedly turns on whether we are dealing explicitly with classification or regression. The development of machine learning methods translates into the selection of algorithms and loss functions. In view of this, substantively the entire article should be improved.
|
||
|
|
Response 4: In this work, we use classification algorithms to predict onion layers and their related metrics, such as classification accuracy and precision. These terms have been described and included in section 4.5 of materials and methods in response to reviewer 2: Classification accuracy (CA) measures the proportion of correctly classified cases out of the total number of cases. Thus, CA = (TP + TN) / (TP + TN + FP + FN), where: TP (True Positives) = Correctly classified positive cases; TN (True Negatives) = Correctly classified negative cases; FP (False Positives) = Negative cases incorrectly classified as positive and FN (False Negatives) = Positive cases incorrectly classified as negative. Precision measures how many of the predicted positive cases are actually correct. Precision = TP / (TP + FP). High accuracy means fewer false positives (FP), indicating that it is more likely to be correct when the model predicts a positive case.
To avoid confusion, we also considered it necessary to change "linear regression" in section 4.5.3 of materials and methods of the manuscript (line 714) to "least squares adjustment".
It is changed in the material and methods sections 4.5 of the manuscript (lines 693-700). We considered it necessary to change "linear regression" in section 4.5.3 of materials and methods of the manuscript (line 732) to "least squares adjustment".
|
||
|
|
Comments 5: When evaluating metrics from a classification point of view, consider precision in addition to recall.
|
||
|
|
Response 5: Thank you for pointing this out. Five metrics associated with the classification models were calculated: 1) AUC (Area Under the Curve) → Area under the curve (usually referred to as the ROC curve, "Receiver Operating Characteristic"); 2) CA (Classification Accuracy) → Classification accuracy (proportion of correctly classified instances); 3) F1 (F1 Score) → F1 score (harmonic mean between precision and recall); 4) Precision → Precision (proportion of predicted positive cases that are actually positive) and 5) Recall (Sensitivity or True Positive Rate, TPR) → Sensitivity or true positive rate (proportion of correctly identified positive cases). Among all of them, the most important ones for our work were chosen (CA and Precision). Specifically, the choice between Precision and Recall in a Machine Learning problem depends on the context of the problem and the consequences of the errors. Using Precision instead of Recall is appropriate when the main goal is to minimize false positives and ensure that positive predictions are reliable, as is our case. We thank the reviewer for this comment and, as detailed in the previous comment, we include the definition of precision and Classification Accurate in materials and methods.
It is changed in the material and methods sections 4.5 of the manuscript (lines 693-700). |
||
|
|
|
||
|
|
Comments 6: Line 555: use the dimensional tolerance for values this figure unacceptable. This also applies to any measurement taken in the research.
|
||
|
|
Response 6: Thank you for this observation. We think that this refers to lines 551-552, which indicate the size of epidermis selected for Tripan Blue staining and subsequent measurements under the microscope. We agree with the reviewer and clarify that squares of epidermis measuring between 4-5 mm on each side were taken for staining, but this size has no influence on subsequent microscopic observation. It is changed in the 4.2 sections of materials and methods of the manuscript: A 4-5 mm square piece (line 551-552)
|
||
|
Comments 7: As part of the data on the design of machine learning models, very little has been described by the authors. Libraries are required, as well as relevant literature from which practical knowledge has been derived.
|
|||
|
Response 7: Thank you for pointing this out. All algorithms and analysis libraries used in the machine learning models are included in Orange3 software [85], programmed in Python, and applying Python scripts. Standard classification models were tested as indicated in section 4.5 of materials and methods (Logistic Regressions, Support Vector Machines (SVM), k-Nearest Neighbors (kNN), Decision Tree, Random Forest, Gradient Boosting, and Neural Networks).
[85] Demšar, J.; Curk, T.; Erjavec, A.; Gorup, Č.; Hočevar, T.; Milutinovič, M.; Možina, M.; Polajnar, M.; Toplak, M.; Starič, A.; Štajdohar, M.; Umek, L.; Žagar, L.; Žbontar, J.; Žitnik, M.; Zupan, B. Orange: Data Mining Toolbox in Python. J. Mach. Learn. Res. 2013, 14, 2349−2353. https://jmlr.csail.mit.edu/papers/v14/demsar13a.html
We do not consider it necessary to change the text in the manuscript.
|
|||
|
|
|||
|
Comments 8: No key action what did the dataset look like for this problem - how many cases, what input variables, what output variables were included in the collection?
|
|||
|
Response 8: We appreciate the comment. The number of biological and technical replicates used in this article is included in the protocols included in the materials and methods section. Regarding the databases used, we appreciate the reviewer's comment and proceed to describe the databases used in the materials and methods section. Thus, the database is composed of 5 datasets: 1) Cellular morphology dataset (cell size and cell number; S1-S6 layers; B, M and U zone); 2) FTIR dataset (absorbance; S1, S2 and S6); 3) CWC dataset (Total sugars, Uronic acids (pectins), Total insoluble sugars (α-cellulose) and Reducing sugars; S1, S2 and S6) 4) NUTC dataset (Total antioxidants, Total phenols, Flavonoids and Total Proteins; S1, S2 and S6) and 5) Gene expression dataset (enzymes Expansin, Fucosidase, Galactosidase, Glucanase, Pectin methylesterase (PME), Pectin/pectate lyase-like (PLL), Polygalacturonase (PG), Xylanase, Xyloglucan endotransglycosylase/hydrolase (XTH) and Xylosidase; S1, S2 and S6). Because this work must be defended as a doctoral thesis, we cannot publish the data openly (e.g. Zenodo or GitHub) since it is confidential. It is changed in the material and methods sections 4.5 of the manuscript (line 677-684).
|
|||
|
Comments 9: What did the test and training set look like?
|
|||
|
Response 9: We understand that this comment is the same as comment 10, so we respond below.
|
|||
|
Comments 10: What proportions were used between the test and training set?
|
|||
|
Response 10: Thanks to the reviewer for pointing out that we didn't detail the percentages. The training/test split was 70% to 30%, which is usually the standard. It is changed in the material and methods sections 4.5 of the manuscript (line 691-692).
|
|||
|
Comments 11: Was data normalisation applied? |
|||
|
Response 11: Thank you for pointing this out. The data were normalized, as indicated in materials and methods: 4.3.1: All spectra (FTIR) were baseline-corrected and area-normalized (800–1800 cm–1) using Omnic software (line 575); 4.4.1. CWE gene selection: A housekeeping gene (Actin) was also chosen to allow gene expression to be normalized in the RT-qPCR assays (line 651) and 4.5.3. Correlational Analysis (line 711). We do not consider it necessary to change the text in the manuscript.
|
|||
|
|
|||
|
Comments 12: In conclusion, the topic should necessarily be improved. It is not good for the research question posed.
|
|||
|
Response 12: Thank you for pointing this out. I hope that with the modifications and clarifications made previously, the understanding of the topic will be improved. We appreciate all the reviewer's comments to improve the manuscript. We do not consider it necessary to change the text in the manuscript.
|
|||
|
Comments 13: The topic is not very much about machine learning I have concerns about the use of these methods in this article. |
|||
|
Response 13: Thank you for pointing this out. We accept your point of view; however, we apply in this work the classification algorithms of Machine Learning to be able to solve a biological problem. Thus, we understand that ML is a very useful and innovative analysis tool that allows obtaining conclusions from the data that describe the growth of onion tissues, mediated by the action of CWE on CWC and NUTC. Likewise, it has been used in other applications that have been previously published such as: 1) Anastomotic leak in colorectal cancer surgery: Contribution of gut microbiota and prediction approaches Hernández‐González et al., 2023. Colorectal Disease 25 (11), 2187-2197; 2) Machine learning study in Caries markers in oral microbiota from Monozygotic Twin Children. Alia-García et al., 2021. Diagnostics 11 (5), 835 We do not consider it necessary to change the text in the manuscript.
|
|||
|
Comments 14: The confusion matrix should specify on which set the results are interpreted.
|
|||
|
Response 14: Thank you for pointing this out. Indeed, the dataset to which the confusion matrix refers (FTIR dataset) is missing from the footer of Figure 3. The description of the databases used has been included in the materials and methods section (as indicated in response to comment 8).
The following is added: (FTIR dataset) in the figure legend of Figure 3 (line 272) of the manuscript. |
|||
|
|
|||
|
Comments 15: Dataset is missing. Dataset should be posted e.g. Mendeley Data or similar repository for data access.
|
|||
|
Response 15: We understand the comment. Regarding the databases used, we appreciate the reviewer's comment and proceed to describe the databases used in the materials and methods section. Thus, we get 5 data sets as described in response to comment 8. Because this work must be defended as a doctoral thesis, we cannot publish the data openly (e.g. Zenodo or GitHub) since it is confidential.
|
|||
|
4. Response to Comments on the Quality of English Language |
|||
|
Point 1: The English could be improved to more clearly express the research. |
|||
|
Response 1: We appreciate the comment. The manuscript has been revised, and several English expressions have been improved. Lines 65, 83, 116 and 151.
|
|||
|
5. Additional clarifications |
|||
|
Again, thank you very much for taking the time to review this manuscript. |
|||

Round 2
Reviewer 2 Report
Comments and Suggestions for Authors
I am satisfied with the revision. Thank you very much.